*Proc. R. Soc. B* **286**: 20190830.

developmental biology, evolution, genomics

marsupial, blastocyst, preimplantation, gene duplication, transcription factor, DDC

**Author for correspondence:**
Peter W. H. Holland
e-mail: peter.holland@zoo.ox.ac.uk

# Of eyes and embryos: subfunctionalization of the *CRX* homeobox gene in mammalian evolution

Amy H. Royall[1], Stephen Frankenberg[2], Andrew J. Pask[2] and Peter W. H. Holland[1]

[1]Department of Zoology, University of Oxford, 11a Mansfield Road, Oxford OX1 3SZ, UK
[2]School of BioSciences, University of Melbourne, Melbourne, 3010 Victoria, Australia

AJP, 0000-0002-1900-2263; PWHH, 0000-0003-1533-9376

ETCHbox genes are fast-evolving homeobox genes present only in eutherian (placental) mammals which originated by duplication and divergence from a conserved homeobox gene, *Cone-rod homeobox* (*CRX*). While expression and function of *CRX* are restricted to the retina in eutherian mammals, ETCHbox gene expression is specific to preimplantation embryos. This dramatic difference could reflect the acquisition of new functions by duplicated genes or subfunctionalization of pleiotropic roles between *CRX* and ETCHbox genes. To resolve between these hypotheses, we compared expression, sequence and inferred function between *CRX* of metatherian (marsupial) mammals and ETCHbox genes of eutherians. We find the metatherian *CRX* homeobox gene is expressed in early embryos and in eyes, unlike eutherian *CRX*, and distinct amino acid substitutions were fixed in the metatherian and eutherian evolutionary lineages consistent with altered transcription factor specificity. We find that metatherian *CRX* is capable of regulating embryonically expressed genes in cultured cells in a comparable way to eutherian ETCHbox. The data are consistent with *CRX* having a dual role in eyes and embryos of metatherians, providing an early embryonic function comparable to that of eutherian ETCHbox genes; we propose that subfunctionalization of pleiotropic functions occurred after gene duplication along the placental lineage, followed by functional elaboration.

## 1. Introduction

Gene duplication is postulated to facilitate evolutionary change and innovation through providing new genetic material which can gain or lose functions over evolutionary time. Although most homeobox gene families are highly conserved across Metazoa, there are examples of homeobox genes that have undergone duplication followed by extreme sequence divergence in some lineages.

The Otx homeobox gene family provides a striking example. Following characterization of the *Drosophila orthodentical* (*otd*) gene, two homologues were rapidly identified in mammals: *OTX1* and *OTX2* [1]. Subsequently, a third gene family member was identified, named *OTX5* or *CRX* [2,3]. These three Otx family genes are descendants of the whole-genome duplications that occurred in vertebrate ancestry; they are found across jawed vertebrates and share high sequence similarity. More recently, it was shown that a series of highly divergent homeobox genes, specific to eutherian mammals, originated later in evolution by tandem duplication from the *CRX* gene followed by extreme sequence divergence away from a canonical Otx-type sequence. Their pattern of sequence divergence has been described as asymmetric, meaning that the 'parental' gene (*CRX*) has undergone far less sequence change than the 'daughter' genes [3–6]. The descendent genes are so distinct in sequence

from Otx family genes that they were given their own gene family names: *Tprx*, *Dprx*, *Leutx* and *Argfx* [7] and collectively ETCHbox [5]. This innovation and sequence change is of note as lineage-specific genes could have important roles in evolution [8]. In eutherian mammals, the 'parental' *CRX* gene locus is flanked by *Tprx1* (called *Crxos* in mouse) and *Tprx2* (Obox genes in mouse); the other duplicates are more distantly located or on another chromosome [5,9]. There is also variation between eutherian species as to which ETCHbox are retained or lost, and which have been duplicated further [5].

Divergent sequence evolution is mirrored by distinct expression patterns and probable functions. The ETCHbox genes of human and cow are expressed predominantly in early development, in a sharp pulse around the 8-cell to 16-cell stages [5,10,11]. Mouse ETCHbox genes are more variable in expression, but are also specific to the very early embryo [9,12,13]. Ectopic expression in cell culture, followed by RNA sequencing (RNA-seq), has uncovered putative roles in controlling a suite of downstream genes, including genes peaking in expression before blastocsyst and implicated in cell fate and fetal–maternal interactions [5,9].

By contrast, the *CRX* gene encodes a transcription factor implicated in the specification of photoreceptor fate in the developing eye [3,14,15]. There is also some evidence for trace expression of *CRX* in the early embryo in human and mouse, but the levels are too low to be identified by whole embryo RNA-seq [5,9,16]. Mouse models with perturbed *CRX* function or *CRX* gene deletion are characterized by the near or complete absence of vision from birth, and mutations in humans can cause cone–rod dystrophy [14,17,18]. Expression in the eye has also been reported for *Crx/Otx5* in frog, chicken, zebrafish (two genes) and dogfish [2,19–21]. However, in some non-mammalian vertebrates, *CRX* expression has also been found in embryonic tissues. For example, in the frog *Xenopus laevis*, *crx* (*Xotx5*) RNA is detected in the Spemann organizer of the gastrula and in anterior neuroectoderm [2,22]; in dogfish *Scyliorhinus canicula*, *Crx* (*Otx5*) expression can be detected in the embryonic brain by *in situ* hybridization in addition to the developing eye [21,23].

It is unclear whether the vertebrate *CRX* gene had ancestral roles in the early embryo, which have been secondarily lost in the eutherian mammal lineage [3], or alternatively whether some sites of embryonic expression were independently gained in *Xenopus* and dogfish. This question is closely tied to understanding the evolution of ETCHbox genes, because these embryo-specific genes were derived from *CRX* in eutherian mammal evolution. If *CRX* was pleiotropic with roles in embryos and eyes prior to the emergence of eutherian mammals, the distinct functions in eutherians could reflect subfunctionalization after gene duplication, followed by gradual and progressive optimization of multiple ETCHbox roles to subtly different embryonic roles [9]. By contrast, if *CRX* was ancestrally eye-specific, ETCHbox genes and their functions would be very radical innovations acquired by eutherian mammals only, thus marking a clear distinction to other vertebrates. It is not clear which scenario is more likely, since eutherian mammal embryogenesis has a mixture of plesiomorphic and derived features. The eutherian mammal lineage encompasses the emergence of the extant Placentalia, characterized by extended embryogenesis and development of a highly invasive placenta that contributes to embryo nutrient

exchange throughout embryogenesis. A comparison to non-eutherian mammals is key to resolving these questions. Marsupials (metatherians) are the immediate outgroup to eutherian mammals, and their embryos do establish maternal interactions albeit with a less invasive placenta than in eutherian mammals [24]. Furthermore, metatherians do not possess ETCHbox genes, as these arose from *CRX* specifically on the eutherian stem lineage.

In this study, we ask whether a metatherian *CRX* gene is eye-specific or whether it also has expression and possible function in the early embryo, comparable to ETCHbox genes. Using sequence comparisons, we first show that specific amino acid changes occurred in the *CRX* proteins of metatherians and eutherians, compatible with alterations to transcription factor function during mammalian evolution. We examined the expression of the *CRX* gene in metatherian development, detecting expression in early embryos and in the eye. To enable comparisons to previously characterized functions of eutherian ETCHbox genes, we expressed metatherian and eutherian *CRX* genes ectopically in cell culture and examined transcriptomic responses using RNA-seq and qPCR. These experiments uncovered similarities in activity between metatherian *CRX* and eutherian ETCHbox genes, consistent with subfunctionalization and progressive specialization after gene duplication in eutherian mammal evolution.

## 2. Material and methods

### (a) Protein sequence analysis

Deduced protein sequences were gathered from NCBI and aligned using MAFFT to identify amino acid substitutions in the homeodomain. The inferred vertebrate ancestral sequence was estimated using FastML [25] (electronic supplementary material, S1A). We tested for positive selection using CodeML implemented in PAML [26], using a branch-model to estimate the dN/dS ratio by assigning two independent ratios, specifying the branch leading to eutherian mammals (model = 2, NSsites = 0).

### (b) Gene expression analysis

To examine mouse gene expression, publicly available RNA-seq datasets (electronic supplementary material, S2A) from across preimplantation development were analysed using kallisto v0.42.4 using GRCm38 coding sequences and default settings to obtain a number of transcripts-per-million mapped reads (TPM) [27]. Coding sequences for *Oboxa7*, *Oboxb2* and *Oboxd2* were manually added to the GRCm38 coding sequence dataset as they are currently unannotated. Other mouse transcripts examined were *Crx* (ENSMUST00000044434.12), *Crxos* (ENSMUST00000171280.2), *Oboxa1* (ENSMUST00000108513.4), *Oboxa4* (ENSMUST00000067288.14), *Otx1* (ENSMUST00000006071.13) and *Otx2* (ENSMUST00000226501.1).

### (c) Dunnart conceptus expression

Dunnart single-cell RNA-seq data were derived from a larger study to be published in a separate manuscript. Dissociated single cells from a range of conceptus stages were processed using Clontech SMART-Seq v4 3′ DE kit, with sample volumes scaled down five-fold. Libraries were prepared using the Illumina Nextera XT DNA Sample Preparation Kit and sequenced at 1 million reads per cell (12 cells per multiplexed library) on an Illumina Nextseq 500 platform. Using the Galaxy Web-based analysis platform (https://usegalaxy.org/), de-multiplexed reads were mapped to dunnart genomic scaffolds (S.F. & A.J.P. 2019, unpublished) using Bowtie for Illumina (default parameters) and

Proc. R. Soc. B 286: 20190830

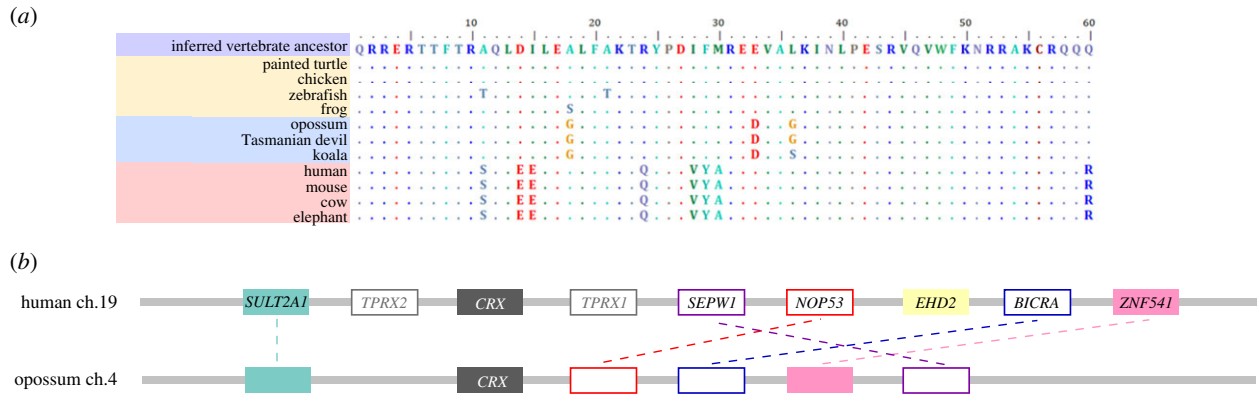

**Figure 1.** (a) Eight amino acid changes are present in all eutherian mammal CRX homeodomains and absent in non-eutherian vertebrates. Two conserved amino acid changes are metatherian-specific and not found outside of metatherians. (b) Synteny of opossum *CRX* relative to orthologous human genomic region confirms the identity of opossum *CRX* locus. Assembly of dunnart genome is not complete enough to describe the syntenic region. (Online version in colour.)

assembled into transcripts using Stringtie and Stringtie Merge. Stringtie was then reapplied to the mapped reads using the Stringtie Merge output as a transcripts reference file to produce the TPM (transcripts per million) dataset reported here. *CRX* was identified among the Stringtie Merge assembled transcripts using BLASTn.

## (d) Reverse transcriptase polymerase chain reaction

Fat-tailed dunnart tissue collections were approved by the University of Melbourne Animal Experimentation Ethics Committees and were in accordance with the National Health and Medical Research Council of Australia (2004) guidelines. Total RNA was extracted from frozen adult female dunnart tissues using the RNeasy Mini Kit (Qiagen), followed by treatment with DNA-*free* (Ambion) to eliminate genomic DNA. Oligo(dt)-primed first-strand cDNA was reverse transcribed using Superscript III (Invitrogen). PCR was performed using GoTaq Green (Promega) in a 20-µl reaction volume for 40 cycles of 95°C for 15 s, 61°C for 20 s and 72°C for 20 s. Dunnart-specific primer sequences were: *CRX*- forward AAGATCAATCTCCCAGAATCCCGAG; *CRX*-reverse AGTC AGTGCATAGGAAGAGGAGG; *TBP*- forward TCCCCAAT-GACCCCTATGACTC; *TBP*-reverse TTTCTGGCTGCTAA CCTGGAC. Products were analysed by electrophoresis.

## (e) Ectopic expression and transcriptomic responses

Plasmids containing V5-tagged human *CRX*, V5-tagged opossum *CRX* (RefSeq: XM_001372308.2, Genscript CloneID: Omc23124) or no homeobox gene, along with a puromycin resistance gene (Oxford Genetics; OG3422), were electroporated using a NEPA21 system (NEPAGENE) into Mouse Embryonic Fibroblasts (MEFs, ScienCell: M7540-57) and allowed to recover for 48 h in DMEM (Gibco: 11965092; 1% PenStrep (Gibco: 15140122), 10% FBS). Cells for immunocytochemistry were fixed in 4% formaldehyde in PBS for twenty minutes before permeabilization using 0.1% Triton-X in 1× PBS, 0.1% Tween-20. Antibodies and visualization are described previously [9].

Cells for RNA harvesting had media replaced for DMEM containing puromycin (0.5 µg ml$^{-1}$). Puromycin was kept in the media for 5 days with a media change on day 3. Puromycin was washed off using multiple short (5 min) incubations in DMEM and cultured overnight. RNA was harvested in triplicate per sample (control, metatherian *CRX*, eutherian *CRX*) using RNeasy Micro Plus kit (Qiagen) following manufacturer's instructions. RNA was prepared for sequencing using the Illumina TruSeq library preparation kit and size of the fragments checked (Agilent D1000 ScreenTape). RNA was sequenced using the Illumina NextSeq500 platform using 2 × 75 bp paired

end reads giving between 68 and 121 million reads per sample. Transcriptome reads were aligned used STAR aligner v2.4.0 (genome assembly: mm10 GRCm38) and Cufflinks v2.2.1 software used to deduce FPKM values; FeatureCounts v1.4.6 produced raw read counts [9]. Clustering using Mfuzz software [28] and gene list overlap analysis was performed as described in Royall *et al.* [9]. Quantitative PCR (qPCR) experiments used New England BioScience Luna Universal One-Step qPCR kit according to manufacturer's instructions and Prime Pro 48 Real-Time qPCR system (Techne). qPCR analysis was performed relative to the *Gapdh* housekeeping gene and relative expression was calculated by comparison to control MEFs transfected with the 'no homeobox gene' plasmid.

Mouse-specific qPCR primers are *Gapdh*-forward CGGGTT CCTATAAATACGGACTG, *Gapdh*-reverse AATACGGCCAAAT CCGTTCA, *Hbegf*-forward TCCCTCTTGCAAATGCCTC, *Hbeg-f*-reverse GGACGACAGTACTACAGCCA, *Ptgs2*-forward TGGG TGTGAAGGGAAATAAGG, *Ptgs2*-reverse GAAGTGCTGGGC AAAGAATG, *Smarca1*-forward TATGCCCTTGAAAGCAGACC, *Smarca1*-reverse TGTTGGAGACTTCTGTGCTG.

To enable comparisons between transcriptomic responses following *CRX* and ETCHbox ectopic expression, differential expression analysis was carried out on FPKM counts using R package DESeq2 [29]. The resulting list of differentially expressed genes was compared with those differentially expressed following ETCHbox ectopic expression (previously obtained using the same methodology as in [9]) and overlapping responses calculated as a percentage of total differentially regulated genes in this study. The extent of overlap was assessed using Fisher's exact test. The GO term enrichment analysis used the functional annotation chart from DAVID Bioinformatics Resources v6.8 [30,31].

# 3. Results

## (a) Eutherian-specific substitutions in CRX homeodomain

We hypothesized that if a pleiotropic CRX protein underwent subfunctionalization during the evolution of eutherian mammals, with some functions partitioned to ETCHbox genes, this may have resulted in novel amino acid substitutions associated with CRX specialization in the eutherian lineage. We searched for *CRX* sequences across vertebrates and confirmed the identity of opossum *CRX* using synteny to mouse and human loci (figure 1). As *CRX* encodes a transcription factor, we focused on the homeodomain sequence responsible for sequence-specific DNA binding and target gene specificity.

Comparison of the CRX homeodomain sequence from a range of vertebrates revealed high conservation, but with substitutions in some lineages (figure 1a).

We find substitutions at eight amino acid sites in the CRX homeodomain of eutherian mammals, all completely conserved between eutherians examined (figure 1a). By contrast, metatherian CRX has undergone substitutions at three sites (two conserved); other lineages have undergone limited change. A phylogenetic tree of CRX protein sequences clearly separates metatherian and eutherian *CRX* genes (electronic supplementary material, S2B).

We tested if the eutherian-specific substitutions were driven by positive selection, as indicated by an elevated dN/dS ($\omega$) ratio, but found the long divergence times were associated with saturation of dS and gave unreliable $\omega$ estimation. Nonetheless, the amino acid changes in eutherian mammals are consistent with an altered function compared to other vertebrates, compatible with the subfunctionalization hypothesis. We find no evidence for reciprocal compensatory changes in ETCHbox homeodomains (electronic supplementary material, S2C).

## (b) *CRX* expression in metatherian embryos and eyes

If subfunctionalization occurred after gene duplication, with retinal functions retained by eutherian *CRX* and embryonic roles retained by diverging ETCHbox genes, then the ancestral mammalian *CRX* gene would have had expression in eyes and in early embryos. If neofunctionalization occurred, with ETCHbox genes acquiring novel roles, the ancestral gene would be eye specific. To resolve between these alternatives we examined *CRX* gene expression in a metatherian.

We analysed single-cell RNA-seq data from early developmental stages of the fat-tailed dunnart (*Sminthopsis crassicaudata*), a marsupial in the family Dasyuridae. We found expression of *CRX* in the early embryo (figure 2), with mean TPM of 79.6 at 16-cell, 14.1 at 32-cell and 40.8 at 64-cell (electronic supplementary material, S1B). By contrast, comparable analysis of mouse and human whole embryo RNA-seq data revealed no expression in the blastocyst or earlier stages, when ETCHbox genes are specifically expressed [5] (figure 2). Using reverse transcriptase polymerase chain reaction (RT-PCR) we confirmed the expected, widely conserved, expression of *CRX* in the adult metatherian eye (electronic supplementary material, S2D). These data imply that the ancestral mammalian *CRX* gene was expressed in embryos and eyes, and in the eutherian lineage these properties were separated after gene duplication.

## (c) Radical restructuring of transcriptomes by the metatherian CRX protein

We hypothesized that a homeodomain protein with diverse roles (in early embryo and eye) would be capable of regulating more target genes and influencing more regulatory networks than a homeodomain protein of highly specialized function. We expressed ectopically a *CRX* gene from a metatherian (*Monodelphis domestica*) and an eutherian mammal (*Homo sapiens*) in primary mouse embryonic fibroblast cells and used RNA-seq to determine effects on global transcriptome, relative to cells transfected with a control vector. Although this experiment does not mimic closely

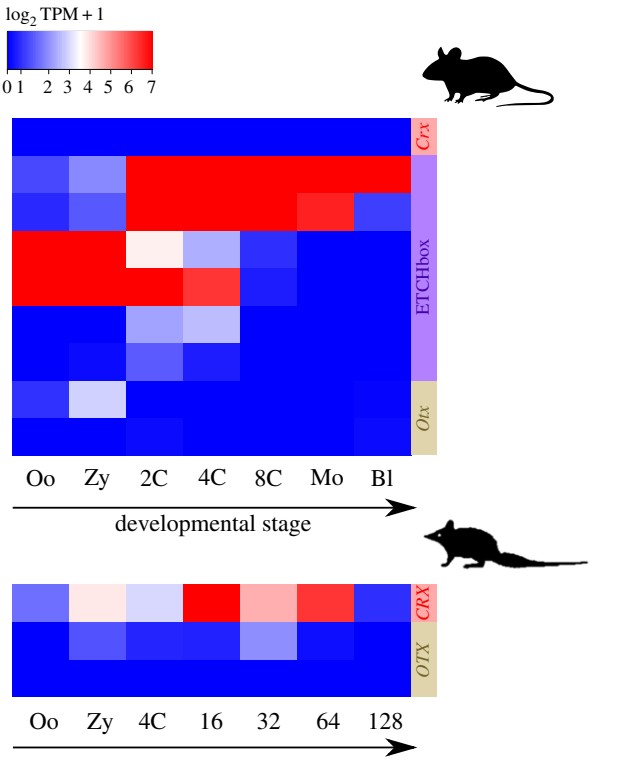

**Figure 2.** Comparison of eutherian (mouse, top) and metatherian (fat-tailed dunnart, bottom) *CRX* gene expression. Mouse ETCHbox genes, but not *Crx*, are expressed in early embryos. Dunnart *CRX* is expressed in early embryos. Expression levels represented as Log2 TPM + 1 (TPM in electronic supplementary material, S1B) Embryonic stages: Oo = oocyte, Zy = zygote, 2C = two cell, 4C = four cell, 8C = eight cell, Mo = morula, Bl = blastocyst, 16 = 16 cell, 32 = 32 cell, 64 = 64 cell, 128 = 128 cell. Embryonic stages not necessarily equivalent between species. ETCHbox genes from top to bottom: *Crxos*, *Oboxa1*, *Oboxa4*, *Oboxa7*, *Oboxb2*, *Oboxd2*. Otx genes from top to bottom: *Otx1*, *Otx2*.

the *in vivo* situation, previous studies have found that ectopic expression of related homeobox genes in cultured cells can drive biologically relevant transcriptomic changes [5,9]. Immunocytochemistry confirmed that both transfected genes are translated into protein (electronic supplementary material, S2E). Following overexpression, the two *CRX* orthologues had different global effects (figure 3). Ectopic expression of metatherian *CRX* resulted in differential expression of 1496 genes (661 up, 835 down; FPKM > 2, $p <$ 0.05; electronic supplementary material, S1C). Ectopic expression of eutherian *CRX* resulted in differential expression of only 580 genes (118 up, 462 down; electronic supplementary material, S1C). There was partial response overlap with 61 common genes upregulated in the two experimental conditions and 278 downregulated. The discrepancy in the number of genes differentially regulated is not due to different levels of ectopic gene expression (electronic supplementary material, S2F) or variations in methodology, as samples were prepared in pairs and associated in the analysis. Gene Ontology (GO) term enrichment analysis showed that genes downregulated following eutherian CRX expression are enriched for cell cycle functions (electronic supplementary material, S1D), whereas those downregulated following metatherian CRX expression are enriched for terms relating to the extracellular matrix (electronic supplementary material, S1E). We conclude that a metatherian CRX

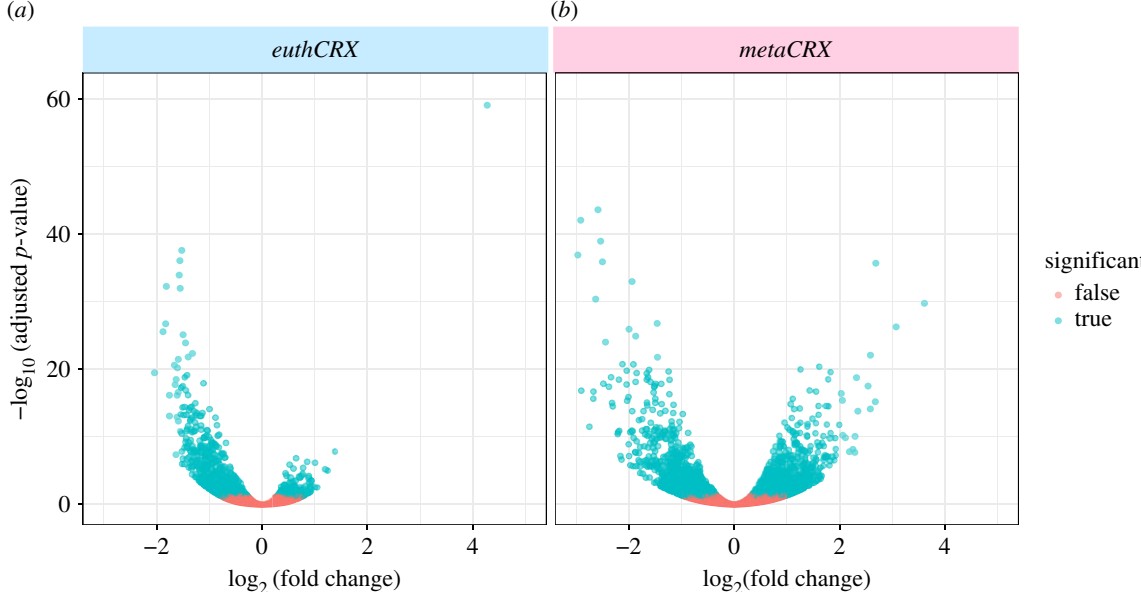

**Figure 3.** Effect on transcriptome of *CRX* ectopic gene expression. Metatherian *CRX* (*metaCRX*; (*b*)) induces more changes in gene expression than eutherian *CRX* (*euthCRX*; (*a*)) following ectopic expression. Both genes have more downregulatory effects than upregulatory effects. Volcano plots of −log10(adjusted *p*-value) and log2 fold-change following differential gene expression analysis. (Online version in colour.)

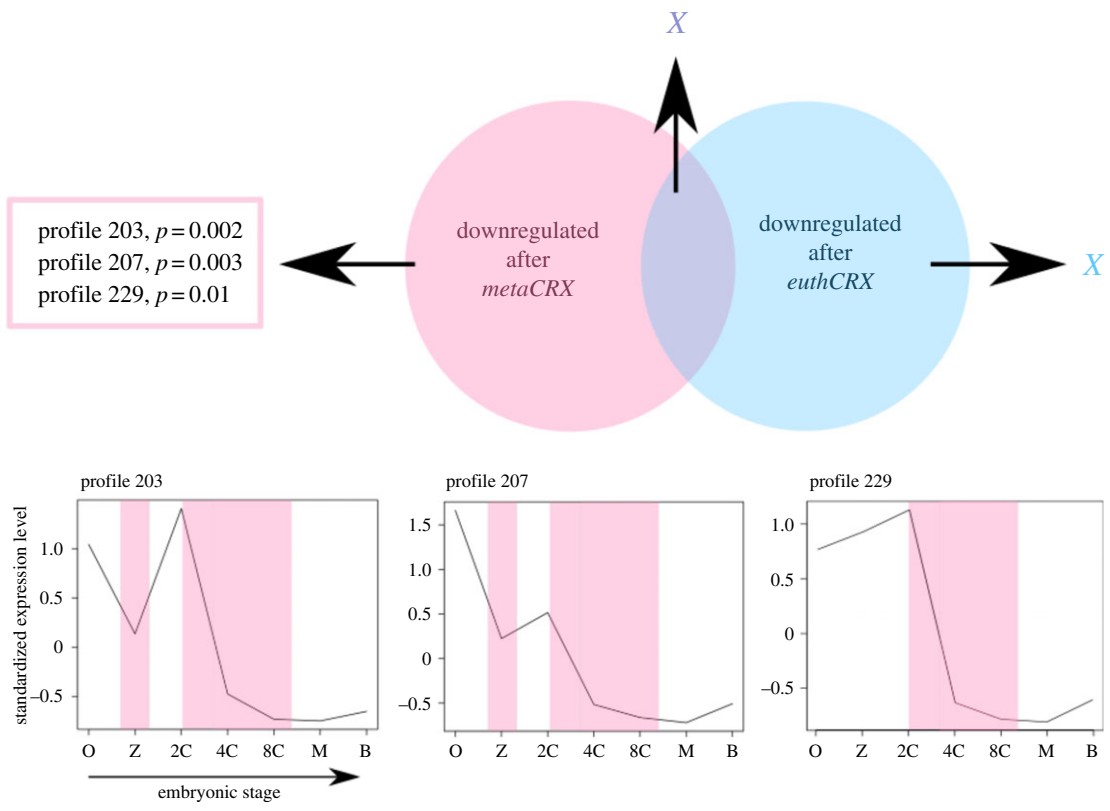

**Figure 4.** Mouse preimplantation clusters significantly regulated by metatherian *CRX* ectopic expression. Coloured strips identify common features of the clusters which we propose are related to the endogenous action of metatherian *CRX*. (Online version in colour.)

homeodomain protein elicits more downstream changes after ectopic expression than a eutherian CRX protein in a mouse fibroblast context.

## (d) Metatherian CRX protein regulates genes expressed in preimplantation development

To assess if transcriptomic changes elicited by CRX protein expression are associated with developmental functions, we first asked how many of the differentially regulated genes

are normally expressed in mouse embryos. We compiled a database of genes expressed in preimplantation mouse embryos [9] and found the transcriptomic response to metatherian *CRX* includes 732 genes that are dynamically expressed in normal preimplantation mouse development (349 up, 383 down, electronic supplementary material, S1F). Of the genes affected by ectopic eutherian *CRX*, only 277 are dynamically expressed in mouse preimplantation development (52 up, 225 down, electronic supplementary material, S1F).

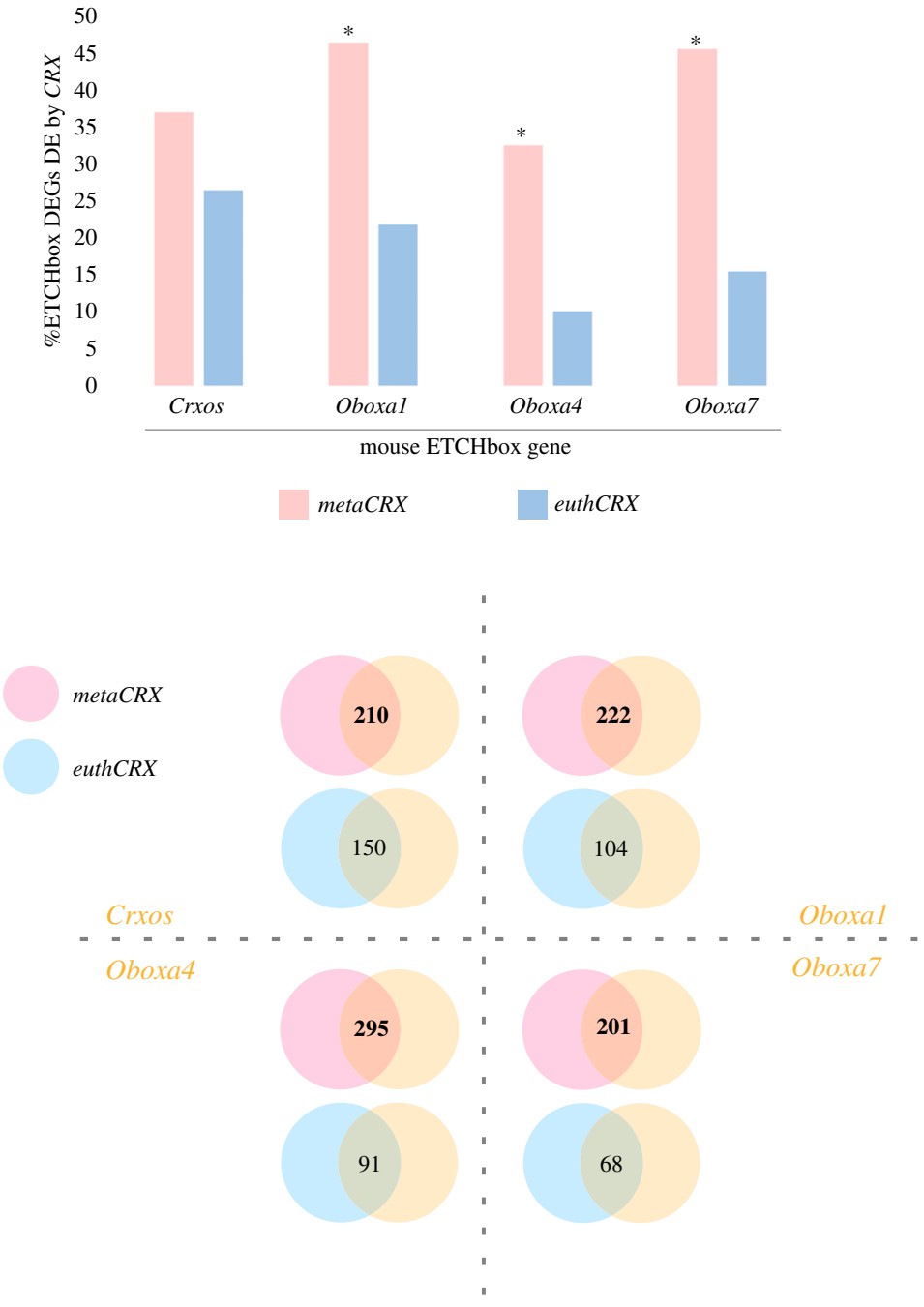

**Figure 5.** Gene sets downstream of mouse ETCHbox genes are more similar to gene sets downstream of metatherian *CRX* than eutherian *CRX*. Comparing lists of differentially regulated genes shows more genes are commonly regulated by metatherian *CRX* and each ETCHbox gene than by eutherian *CRX* and ETCHbox genes. (*a*) Percentage of DEGs following ETCHbox ectopic expression also regulated following *CRX* ectopic expression. Analyses show that the overlaps between Obox DEGs and *metaCRX* DEGs are stronger than the overlaps between Obox DEGs and *euthCRX* DEGs when the total sample size is taken into account (indicated by *). (*b*) Numbers of genes in the overlap between *metaCRX* and *euthCRX* with four mouse ETCHbox genes are shown by Venn diagrams.

Grouping developmentally expressed genes into sets based on their temporal expression profiles reduces the complexity of analysis and can help distinguish signal from noise [5,9,10]. We asked if the differentially expressed genes were enriched for genes that follow particular temporal profiles in development. If profile enrichment is found, this can increase confidence in the biological relevance of the transcriptomic response and provide clues to putative *in vivo* roles [5,9,10]. We used a clustering approach to assign all mouse genes to 85 different temporal expression profiles spannning early embryonic development (elctronic supplementary material, S2G), and tested if the lists of genes

differentially regulated after *CRX* ectopic expression were enriched for particular temporal profiles.

We did not find profile enrichment in the lists of genes upregulated either by metatherian CRX or by eutherian CRX proteins. We also found no profile enrichment in the set of genes downregulated by eutherian *CRX* ectopic expression, or the common set of genes downregulated by both eutherian and metatherian CRX proteins (figure 4). However, we found significant enrichment for three temporal profiles in the list of genes downregulated by only metatherian *CRX* ectopic expression (profiles 203, 207 and 229; electronic supplementary material, S2G). The temporal gene expression

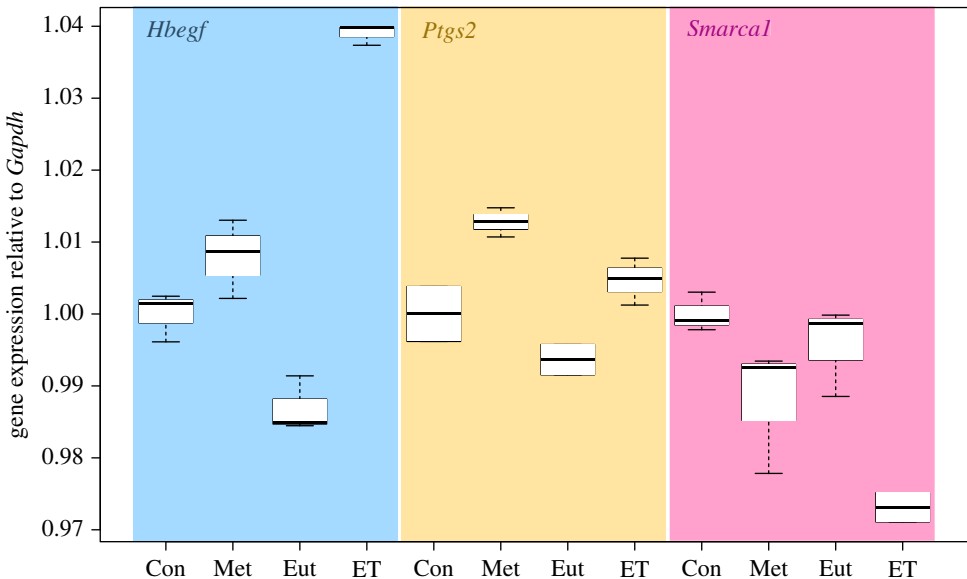

**Figure 6.** Genes with known roles in early eutherian development are similarly regulated following metatherian *CRX* and ETCHbox ectopic expression. A qPCR assay confirms that following metatherian *CRX* or *Oboxa1* ectopic expression, RNA levels of both *Hbegf* and *Ptgs2* are increased, but following eutherian *Crx* ectopic expression neither are upregulated. Similarly, following ETCHbox and metatherian *Crx*, but not eutherian *Crx*, ectopic expression Smarca1 expression decreases. Con = control, Met = metatherian *CRX*, Eut = eutherian *CRX*, ET = ETCHbox gene (*Oboxa1*). (Online version in colour.)

patterns encapsulated by these profiles differ in detail, but all three are composed of mouse genes showing a reduction in gene expression level between the two-cell and eight-cell stages of embryonic development (figure 4). We conclude that the metatherian CRX protein, but not the eutherian CRX protein, is capable of downregulating genes with common expression patterns in the early mouse embryo and is more likely to have *in vivo* functions at this period of development. Specifically, metatherian CRX can regulate a suite of genes that show declining expression in early mouse development. We propose that this activity reflects an *in vivo* role of the metatherian *CRX* gene: downregulating expression of specific genes before blastocyst formation.

## (e) Common downstream effects of metatherian *CRX* and eutherian Obox genes

If the embryonic roles of eutherian ETCHbox genes arose initially by subfunctionalization, a prediction is that metatherian *CRX* should have comparable embryonic roles to ETCHbox. Specifically, we hypothesized that ectopic expression of metatherian CRX protein would elicit similar downstream effects to ectopic expression of ETCHbox genes. As ETCHbox genes subsequently underwent further duplication and diversification they may have gained or lost roles, but some similarities should remain. By contrast, if the embryonic functions of ETCHbox genes are a novelty that arose by neofunctionalization, expression of either metatherian or eutherian *CRX* would have equally dissimilar downstream effects to ETCHbox genes. To test these predictions, we took the lists of embryonic genes differentially regulated by ectopic *CRX* expression and compared these to similarly derived lists for mouse ETCHbox genes [9]. Figure 5a displays these comparisons as the percentage of ETCHbox responsive genes (differentially expressed genes, DEGs) that are also *CRX* responsive. These lists will include direct and indirect targets. Figure 5b shows the data displayed as absolute numbers of shared responsive genes. These comparisons show that, for all ETCHbox genes

examined, the overlap in responsive genes is larger for metatherian *CRX* than for eutherian *CRX*. To test whether the latter conclusion is driven primarily by transcriptomic profile or is an artefact of sample size (eutherian *CRX* expression affects fewer genes globally than does metatherian *CRX*), we used Fisher's exact test. This showed that overlap between metatherian *CRX* and murine Obox genes is stronger than that for eutherian *CRX* even when the sample size is accounted for (figure 5a). We conclude that metatherian *CRX* affects the expression of a similar, but not identical, set of targets as do *Crxos* and Obox genes (mouse ETCHbox genes).

As metatherian *CRX* and mouse ETCHbox genes are capable of eliciting similar transcriptomic effects, we asked what embryonic functions might be the basis of the similarity. We focused on genes with well-characterized roles in early mammalian embryogenesis. Metatherian *CRX* ectopic expression leads to differential expression of genes with reported roles in early eutherian development including maternal communication (*Hbegf*), downregulation of the maternal inflammatory response (*Ptgs2*) and chromatin remodelling (*Smarca1*) among others (electronic supplementary material, S1G). Each of these genes was also regulated by one of more mouse ETCHbox genes, but none are regulated following eutherian *CRX* gene overexpression. Furthermore, we repeated transfections of all eutherian *CRX*, metatherian *CRX* and *Oboxa1* and treated samples in parallel. Transcriptome analysis by qPCR shows that *Hbegf*, *Ptgs2* and *Smarca1* are regulated by metatherian *CRX* and *Oboxa1* in the same direction, emphasizing the similarity of likely embryonic roles. Again, similar regulation following eutherian *CRX* expression is not observed (figure 6).

## 4. Discussion

The duplication, degeneration and complementation (DDC) model of gene evolution postulates that duplication of a pleiotropic gene leads to functional redundancy, followed

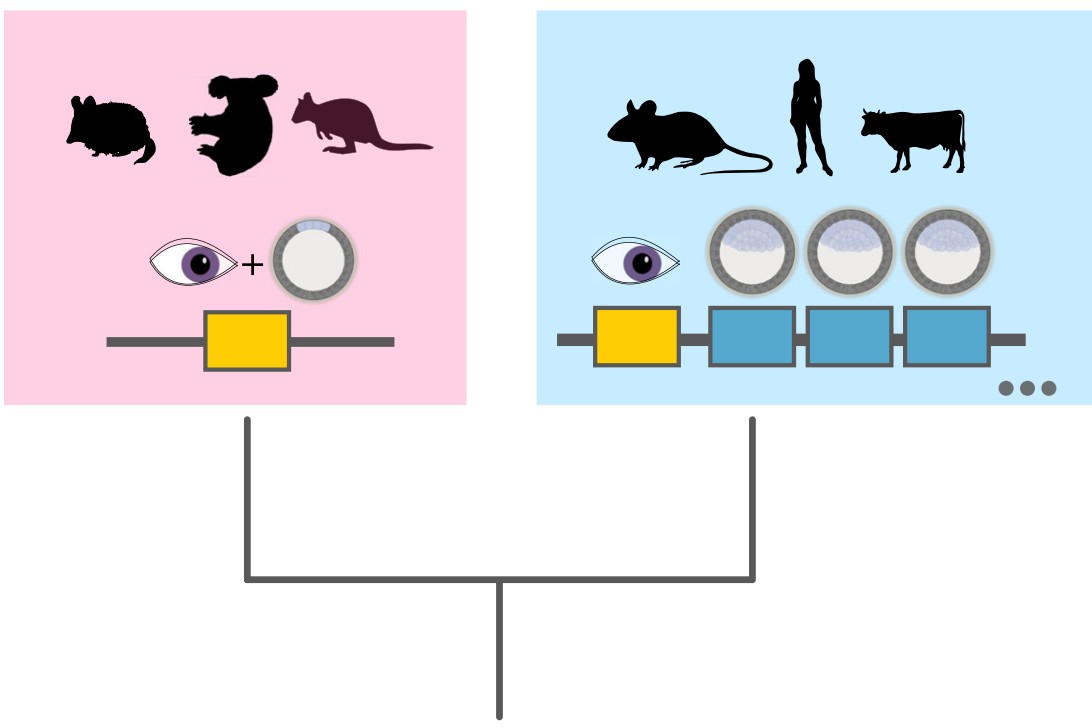

**Figure 7.** Duplication of *CRX* in eutherian lineage leads to specialization of duplicates into embryonic or retina-specific functions. In metatherians, *CRX* (yellow) continues to carry out both functions; in eutherians, embryonic roles are retained and elaborated on by ETCHbox genes (blue) while eutherian *CRX* retains functions in the eye. (Online version in colour.)

by stochastic mutation of different functional elements in each duplicate gene and distribution of ancestral gene functions between the duplicates [32]. This can be an intermediate step in the evolution of new or optimized gene functions, because if a pleiotropic gene is deployed for two or more roles in the same organism, it might not be possible for natural selection to optimize the gene for each role simultaneously. This resolution from compromise has been termed 'escape from adaptive conflict' [33]. Alternatively, when a new gene originates by a duplication event, it could acquire entirely new functions (or additional functions) through 'neofunctionalization'. Distinguishing between these alternatives is crucially important if we wish to understand the origin and evolution of biological traits. However, they are not always straightforward to distinguish as this needs comparable information from outgroups, ideally close outgroups.

There is currently much interest in the processes that underpin early development of humans, from fertilization and embryonic genome activation through to the determination of cell fates, as these events have practical relevance to stem cell biology and assisted reproductive technologies. The ETCHbox genes of humans and other eutherian mammals are expressed after embryonic genome activation and before cell fate determination and may be central to these processes [5,9,11,34]. We wished to understand how these gene functions arose in evolution as, paradoxically, their most closely related and parental gene (*CRX*) is expressed almost exclusively in the eye. Using marsupials (metatherians) as an outgroup, we found evidence that after the duplication of *CRX* in eutherians, subfunctionalization occurred. The *CRX* gene retained an ancestral role in the eye, while the highly divergent ETCHbox genes retained ancestral functions in the early embryo; our data argue that the single *CRX* gene of marsupials has dual properties, aligning with the DDC model (figure 7).

Subfunctionalization can occur through regulatory sequence evolution and/or coding sequence evolution. We find evidence for both. Changes in gene expression are indicative of regulatory evolution and it is striking that metatherian *CRX* is expressed in the very early stages of embryogenesis of the fat-tailed dunnart, as well as in adult eye, whereas in eutherian mammals *CRX* expression is limited to the eye. Coding sequence change is evident in the homeodomain of eutherian CRX, suggesting that sequence-specific DNA binding or transcription factor activity was altered in association with expression restriction to the eye. Changes to the encoded protein sequence clearly have functional relevance because when each is expressed in an ectopic situation, the eutherian and metatherian CRX proteins elicit different downstream effects. These differences could only be caused by protein sequence differences. When expressed ectopically in cultured cells, we found that a metatherian CRX protein caused a significant reduction in expression of sets of genes that would also normally fall in expression in the preimplantation embryo. Expressing eutherian CRX protein did not elicit the same transcriptomic response. These experiments also showed that the transcriptional effect of metatherian *CRX* ectopic expression has similarities to the effect of ETCHbox ectopic expression. For example, the ETCHbox gene *Oboxa1* and the metatherian *CRX* cause the same effect on the expression of candidate genes with known importance in both metatherian and eutherian embryo-maternal communication, *Hbegf* and *Ptgs2* [35], and on components of the SWI/SNF chromatin remodelling factor, *Smarca1*. We propose that metatherian *CRX*, and by extrapolation *CRX* of the common therian ancestor, functioned in the early embryo and the eye; following tandem gene duplication in eutherians, eye-associated functions were retained by one gene and embryonic functions were retained by a larger array of duplicates following

DDC. Paradoxically, these latter duplicates, the ETCHbox genes, diverged greatly in sequence despite apparently retaining some ancestral roles.

Under the classical DDC model, one gene with two functions duplicates and each duplicate takes on one of the functions and becomes specialized. However, in the case studied here, the fate of the duplicated genes has been more complex. The *CRX* gene duplication events in eutherians gave rise to many gene copies. One of the copies, referred to by the same name as the ancestral gene *CRX*, underwent a rather limited degree of sequence change (although this did alter protein function, as discussed above). The other gene copies underwent extreme sequence change, and are barely recognizable as duplicates of *CRX* [5]. These are the genes that inherited the embryonic functions from *CRX* in an ancestral mammal.

A key question is why eutherians deploy multiple *CRX*-derived genes in early development when metatherian development occurs with the expression of a single *CRX* orthologue? And why have these *CRX*-derived genes diverged so radically in protein sequence from the ancestral sequence and from each other? We do not have clear answers to these questions, but they may lie in differences in early embryonic development. For example, eutherians have an extended gestation compared with metatherians, in which the embryo implants into the maternal tissue through an invasive (though quite variable) placenta. This may require regulation of additional gene activities, not present in metatherians. In addition, there are morphological differences between the embryos themselves; metatherian embryos do not have an inner cell mass at the blastocyst stage, instead having embryo-fated cells as a pluriblast on the surface, nor do they undergo cellular compaction [24,36–38]. These distinctions are candidate explanations for asymmetrical gene divergence of *CRX* and ETCHbox genes in eutherians, and for selective retention of multiple duplicates. The variation in copy number between eutherian species might also relate to variation in placental structure,

or it may be an evolutionary consequence of partial redundancy [9]. This reasoning implies that the evolution of these genes in eutherians probably involved more than the DDC mechanisms that we uncover here. In addition to ancestral pleiotropic roles being divided between the descendent genes, we propose that ETCHbox genes acquired additional functions to 'fine-tune' embryo development, and for additional embryo–maternal interactions that were not present in the ancestral *CRX* gene. Therefore, rather than straightforward subfunctionalization under a DDC model, we suggest that ETCHbox genes have secondarily become entwined in regulatory networks specific to eutherian embryos through specialization (figure 7). We term this subfunctionalization-specialization mechanism DDC+.

We conclude that following *CRX* duplication in the evolution of eutherian mammals, aspects of classical DDC-mediated subfunctionalization occurred. However the influence of additional gene duplication events and extreme sequence divergence allowed incorporation of additional functions, thus adding a component of neofunctionalization on top of subfunctionalization and the realization of DDC+.

Data accessibility. Raw sequencing datasets are accessible through the NCBI Gene Expression Omnibus (GSE129456). The mouse developmental stage RNA-seq datasets analysed in this study were obtained from NCBI SRA project SRP034543 [39].

Authors' contributions. A.H.R. and P.W.H.H. conceived and designed the study. A.H.R. performed sequence analyses, molecular and cell biology experiments, transcriptomics and statistical analyses of data. S.F. and A.J.P. carried out dunnart embryo sequencing and analysis. A.H.R. and P.W.H.H. wrote the manuscript and all authors contributed to and approved the final manuscript.

Competing interests. We declare we have no competing interests.

Funding. We acknowledge BBSRC funding to A.H.R. through the Oxford Interdisciplinary Bioscience Doctoral Training Partnership (BB/M011224/1) and Discovery Project funding (DP160103683) from the Australian Research Council to A.J.P. and SF.

Acknowledgements. We thank Rodrigo Pracana, Yichen Dai, Aris Katzourakis, Sounak Sahu, Aziz Aboobaker, Anish Dattani, Sonia Trigueros, Tim Davies and Paul Fairchild for discussions and advice.

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
