## [Reviewer comments · Proceedings of the Royal Society B: Biological Sciences]

Review History

RSPB-2019-0830.R0 (Original submission)

Review form: Reviewer 1

Recommendation

Accept with minor revision (please list in comments)

Scientific importance: Is the manuscript an original and important contribution to its field?

Good

General interest: Is the paper of sufficient general interest?

Good

Quality of the paper: Is the overall quality of the paper suitable?

Good

Is the length of the paper justified?

Yes

Should the paper be seen by a specialist statistical reviewer?

Yes

Do you have any concerns about statistical analyses in this paper? If so, please specify them explicitly in your report.

No

It is a condition of publication that authors make their supporting data, code and materials available - either as supplementary material or hosted in an external repository. Please rate, if applicable, the supporting data on the following criteria.

Is it accessible?

Yes

Is it clear?

Yes

Is it adequate?

Yes

Do you have any ethical concerns with this paper?

No

Comments to the Author

The manuscript by Royall et al. focusses on Crx genes in mammals to address the issue of gene evolution following duplication. As previously shown by the Holland group, this gene family has undergone several duplications followed by asymmetric evolution in the eutherian lineage. The biological significance of these events and the mode of evolution of the paralogues remain unclear. Based on comparisons of expression dynamics during early development and of transcriptomic responses elicited in embryonic fibroblasts by overexpression of a metatherian CRX representative (*M. domestica*) and CRX paralogues from human, the authors suggest that this mode of evolution has involved a novel mechanism, referred to as subfunctionalisation-specialisation.

The question is well presented, the manuscript well written and generally well interpreted. I have, though, suggestions for additional analyses and clarifications:

- 1.- the description of the gene datasets obtained following over-expression of metatherian /eutherian CRX related genes is essentially numerical, with the exception of a few selected candidates. A much more comprehensive analysis, including at least a systematic GO term analysis, would be important to discuss functional evolution
2. the biological significance of the cellular system used should be discussed. The results obtained in this context may not reflect a good picture of the regulatory networks, which CRX/ETCH box genes may control in vivo and "function" in this context has a very specific meaning that should be explicit.
3. along the same line, the authors discuss ETCH box genes functions during early development but in the absence of direct functional data, based for instance of loss-of-function experiments in the mouse, this discussion remains extremely speculative. What is the direct evidence that "eutherians use multiple CRX-derived genes in early development when metatherian development can occur with a single CRX orthologue" (line 336)? The multiple losses undergone by these genes in eutherians should also be discussed. A clearer distinction between hypotheses, their rationale, and demonstrated facts would be needed.

Minor point: there are a lot of typos throughout the main text and Supplementary legends.

Review form: Reviewer 2

Recommendation

Major revision is needed (please make suggestions in comments)

Scientific importance: Is the manuscript an original and important contribution to its field?

Good

General interest: Is the paper of sufficient general interest?

Good

Quality of the paper: Is the overall quality of the paper suitable?

Good

Is the length of the paper justified?

Yes

Should the paper be seen by a specialist statistical reviewer?

No

Do you have any concerns about statistical analyses in this paper? If so, please specify them explicitly in your report.

No

It is a condition of publication that authors make their supporting data, code and materials available - either as supplementary material or hosted in an external repository. Please rate, if applicable, the supporting data on the following criteria.

Is it accessible?

Yes

Is it clear?

Yes

Is it adequate?

Yes

Do you have any ethical concerns with this paper?

No

Comments to the Author

Please address especially this comment:

Lines 333-335,

333 protein function, as discussed above). The other gene copies underwent extreme sequence change, 334 and are barely recognisable as duplicates of CRX 5. These are the genes that inherited the embryonic 335 functions from CRX in ancestral mammal.

Please comment more on this apparent paradox. Metatherian mammals invented viviparity and one would think that whatever gene regulatory networks made such an evolutionary and revolutionary invention possible are likely to be conserved. For a whole other set of genes (albeit derived by duplication) to take over THE SAME invention requires some more persuasion than just "inherited." As is, your description sounds as if viviparity were invented twice, once by metatherians and later by eutherians. (See Appendix A)

Decision letter (RSPB-2019-0830.R0)

20-May-2019

Dear Professor Holland:

Your manuscript has now been peer reviewed and the reviews have been assessed by an Associate Editor. The reviewers' comments (not including confidential comments to the Editor) and the comments from the Associate Editor are included at the end of this email for your reference. As you will see, the reviewers and the Editors have raised some concerns with your manuscript and we would like to invite you to revise your manuscript to address them.

Research ethics:

Use of animals and field studies:

It is a condition of publication that you make available the data and research materials supporting the results in the article. Datasets should be deposited in an appropriate publicly available repository and details of the associated accession number, link or DOI to the datasets must be included in the Data Accessibility section of the article

(<https://royalsociety.org/journals/ethics-policies/data-sharing-mining/>). Reference(s) to datasets should also be included in the reference list of the article with DOIs (where available).

Please submit a copy of your revised paper within three weeks. If we do not hear from you within this time your manuscript will be rejected. If you are unable to meet this deadline please let us know as soon as possible, as we may be able to grant a short extension.

Best wishes,
Professor John R. Hutchinson, Editor
Proceedings B
mailto: proceedingsb@royalsociety.org

Associate Editor

Board Member: 1

Comments to Author:

We have now received two reviews for this manuscript. Both are positive, but both also highlight some conceptual issues that should be addressed before publication. Based on my own reading, I agree with the comments of the reviewers. As such, I suggest that the authors respond to and address all of the concerns raised by the reviewers.

Reviewer(s)' Comments to Author:

Referee: 1

Comments to the Author(s)

The manuscript by Royall et al. focusses on Crx genes in mammals to address the issue of gene evolution following duplication. As previously shown by the Holland group, this gene family has undergone several duplications followed by asymmetric evolution in the eutherian lineage. The biological significance of these events and the mode of evolution of the paralogues remain unclear. Based on comparisons of expression dynamics during early development and of transcriptomic responses elicited in embryonic fibroblasts by overexpression of a metatherian CRX representative (*M. domestica*) and CRX paralogues from human, the authors suggest that this mode of evolution has involved a novel mechanism, referred to as subfunctionalisation-specialisation.

The question is well presented, the manuscript well written and generally well interpreted. I have, though, suggestions for additional analyses and clarifications:

- 1.- the description of the gene datasets obtained following over-expression of metatherian /eutherian CRX related genes is essentially numerical, with the exception of a few selected candidates. A much more comprehensive analysis, including at least a systematic GO term analysis, would be important to discuss functional evolution
2. the biological significance of the cellular system used should be discussed. The results obtained in this context may not reflect a good picture of the regulatory networks, which CRX/ETCH box genes may control in vivo and "function" in this context has a very specific meaning that should be explicit.
3. along the same line, the authors discuss ETCH box genes functions during early development but in the absence of direct functional data, based for instance of loss-of-function experiments in the mouse, this discussion remains extremely speculative. What is the direct evidence that "eutherians use multiple CRX-derived genes in early development when metatherian development can occur with a single CRX orthologue" (line 336)? The multiple losses undergone by these genes in eutherians should also be discussed. A clearer distinction between hypotheses, their rationale, and demonstrated facts would be needed.

Minor point: there are a lot of typos throughout the main text and Supplementary legends.

Referee: 2

Comments to the Author(s)

Please address especially this comment:

Lines 333-335,

333 protein function, as discussed above). The other gene copies underwent extreme sequence change, 334 and are barely recognisable as duplicates of CRX 5. These are the genes that inherited the embryonic 335 functions from CRX in ancestral mammal.

Please comment more on this apparent paradox. Metatherian mammals invented viviparity and one would think that whatever gene regulatory networks made such an evolutionary and revolutionary invention possible are likely to be conserved. For a whole other set of genes (albeit derived by duplication) to take over THE SAME invention requires some more persuasion than just "inherited." As is, your description sounds as if viviparity were invented twice, once by metatherians and later by eutherians.

Author's Response to Decision Letter for (RSPB-2019-0830.R0)

See Appendix B.

RSPB-2019-0830.R1 (Revision)

Review form: Reviewer 1

Recommendation

Accept as is

Scientific importance: Is the manuscript an original and important contribution to its field?

Good

General interest: Is the paper of sufficient general interest?

Good

Quality of the paper: Is the overall quality of the paper suitable?

Good

Is the length of the paper justified?

Yes

Should the paper be seen by a specialist statistical reviewer?

Yes

Do you have any concerns about statistical analyses in this paper? If so, please specify them explicitly in your report.

No

It is a condition of publication that authors make their supporting data, code and materials available - either as supplementary material or hosted in an external repository. Please rate, if applicable, the supporting data on the following criteria.

Is it accessible?

Yes

Is it clear?

Yes

Is it adequate?

Yes

Do you have any ethical concerns with this paper?

No

Comments to the Author

Referee 2 appreciates the authors' responses.

Review form: Reviewer 2

Recommendation

Accept as is

Scientific importance: Is the manuscript an original and important contribution to its field?

Good

General interest: Is the paper of sufficient general interest?

Good

Quality of the paper: Is the overall quality of the paper suitable?

Good

Is the length of the paper justified?

Yes

Should the paper be seen by a specialist statistical reviewer?

No

Do you have any concerns about statistical analyses in this paper? If so, please specify them explicitly in your report.

No

It is a condition of publication that authors make their supporting data, code and materials available - either as supplementary material or hosted in an external repository. Please rate, if applicable, the supporting data on the following criteria.

Is it accessible?

Yes

Is it clear?

Yes

Is it adequate?

Yes

Do you have any ethical concerns with this paper?

No

Comments to the Author

The authors have satisfactorily answered my questions and concerns. The manuscript provides an interesting contribution to the issue of gene evolution following duplication.

Decision letter (RSPB-2019-0830.R1)

28-Jun-2019

Dear Professor Holland

I am pleased to inform you that your manuscript entitled "Of eyes and embryos: subfunctionalisation of the *CRX* homeobox gene in mammalian evolution" has been accepted for publication in Proceedings B. Congratulations!! The reviewers both were fully convinced by the revisions.

Open Access

Paper charges

Sincerely,

Professor John Hutchinson
Editor, Proceedings B
<mailto:proceedingsb@royalsociety.org>

Reviewer(s)' Comments to Author:

Referee: 1

Comments to the Author(s)

Referee 2 appreciates the authors' responses.

Referee: 2

Comments to the Author(s)

The authors have satisfactorily answered my questions and concerns. The manuscript provides an interesting contribution to the issue of gene evolution following duplication.

Appendix A

Review of RSBP-2019-0830

This paper describes the expression patterns of the gene CRX and its derivatives, ETCHbox genes, in two representatives of the mammalian subgroups eutherians (mouse) and metatherians (fat-tailed dunnart). While CRX is expressed in both, it is restricted to the retina in the mouse. In the dunnart, it is additionally expressed in early embryos. Meanwhile, CRX derivatives (the group called ETCH box genes) are expressed in preimplantation mouse embryos and not at all in the dunnart. Dunnart CRX appears functionally equivalent to the mouse ETCHbox and thus performs a dual role in dunnarts. The paper concludes that subfunctionalisation of pleiotropic functions by CRX in eutherian mammals arose after gene duplication.

Line 23,

23 *Drosophila* **orthodentical** (*otd*) gene, two homologues were rapidly identified in mammals: OTX1 and...

Orthodenticle, not orthodentical.

Line 58,

58 specific genes were derived from CRX in eutherian mammal evolution. If CRX **was** pleiotropic with...

Were, not was.

Line 62,

62 if CRX **was** ancestrally eye-specific, ETCHbox genes and their functions would be very radical

Were, not was.

Line 67,

67 a highly invasive placenta that contributes to **embryo nutrient exchange** throughout embryogenesis.

Nutrient exchange does not occur between conceptus and mother; it is a one-way supply from the latter to the former. Gas exchange, yes, but not nutrients.

Lines 69-70,

Marsupials 69 (metatherians) are the immediate outgroup to eutherian mammals, and their embryos do establish 70 maternal interactions, albeit with a **less invasive placenta** than in eutherian mammals

Please elaborate the use of "less invasive." Marsupials form a yolk-sac placenta; placentals form a chorio-allantoic one. However, a few marsupials (Peramelids) form an invasive chorioallantoic one in addition to a yolk-sac placenta. "Less invasive" suggests misleadingly that the placentas are the same in both groups of mammals, but differ only in invasiveness.

Lines 73-83,

73 In this study we **ask** whether a metatherian CRX gene is eye-specific or whether it also has 74 expression and function in the early embryo, comparable to ETCHbox genes. Using sequence 75

comparisons, we first **show** that specific amino acid changes occurred in the CRX proteins of 76 metatherians and eutherians, compatible with alterations to transcription factor function during 77 mammalian evolution. We **examine** expression of the CRX gene in metatherian development, 78 detecting expression in early embryos and in eye. To enable comparisons to previously characterised 79 functions of eutherian ETCHbox genes, we **expressed** metatherian and eutherian CRX genes 80 ectopically in cell culture, and **examined** transcriptomic responses using RNA-seq and QPCR. These 81 experiments **uncovered** functional similarities between metatherian CRX and eutherian ETCHbox 82 genes, indicative of subfunctionalisation and progressive specialisation after gene duplication in 83 eutherian mammal evolution.

Use past tense uniformly to denote that all actions described were completed.

Lines 314-317,

Changes to the encoded protein sequence clearly have 315 functional relevance because when each is expressed in an ectopic situation, the eutherian and 316 metatherian CRX proteins have very different downstream effects. These differences **could only be 317 caused** by protein sequence differences.

Please revise text to indicate that if differences in tertiary structure did arise because of amino-acid-sequence changes, then protein function could (but not with certainty) be also affected. Changes in the primary structure of a protein could well cause “silent” (that is, non-existent functionally) changes (mutations!) if they are not reflected in the protein’s tertiary structure.

Lines 333-335,

333 protein function, as discussed above). The other gene copies underwent extreme sequence change, 334 and are barely recognisable as duplicates of CRX 5. These are the genes that inherited the embryonic 335 functions from CRX in ancestral mammal.

Please comment more on this apparent paradox. Metatherian mammals invented viviparity and one would think that whatever genes made such an evolutionary invention possible are likely to be conserved. For a whole other set of genes (albeit derived by duplication) to take over THE SAME invention requires some more persuasion than just “inherited.” As is, your description sounds as if viviparity were invented twice, once by metatherians and later by eutherians.

Lines 338-341,

338 genes diverged so radically in protein sequence? The answers may lie in differences in early embryonic 339 development. For example, eutherians have an extended gestation compared to metatherians, in 340 which the embryo implants into the maternal tissue through an invasive (though quite variable) 341 placenta. This may require regulation of additional gene activities, not present in metatherians.

Please revise text to clarify that metatherian and eutherian placentas are materially, developmentally, and anatomically different and that they do not differ only in terms of invasiveness. See also comment above on Lines 69-70.

Appendix B

We thank the Editors and Reviewers for their interest in the manuscript and their helpful comments. We have acted on all of these, as detailed below, and believe they have improved the clarity and robustness of the paper. We hope that these changes satisfy the requirements for acceptance.

Referee: 1

Comments to the Author(s): The manuscript by Royall et al. focusses on Crx genes in mammals to address the issue of gene evolution following duplication (deleted)....The question is well presented, the manuscript well written and generally well interpreted. I have, though, suggestions for additional analyses and clarifications.

1.- the description of the gene datasets obtained following over-expression of metatherian /eutherian CRX related genes is essentially numerical, with the exception of a few selected candidates. A much more comprehensive analysis, including at least a systematic GO term analysis, would be important to discuss functional evolution

Our apologies that we were not clear enough in our explanation of analyses. Our analysis of gene datasets is not primarily numerical but makes use of each list of gene names (gene identities) that are up- or down-regulated. We compare the gene datasets affected by ectopic expression with independently generated gene sets grouped by temporal expression profiles in normal development. Thus, rather than grouping genes by function (as in GO), we group genes by embryonic expression pattern. We and others have found this a powerful analysis method when dealing with embryonic genes. Specifically, in this study we show that metatherian CRX expression leads to differential expression of groups of genes that are down-regulated in the eutherian preimplantation embryo (Figure 4), whereas the eutherian CRX dataset does not show the same enrichment. We suggest that this analysis allows us to discuss function more comprehensively than GO term analysis, as we see when the potential target genes are utilised in a developmental context. It is likely that ETCHbox genes, being homeobox genes, are regulating a transition in the developmental progression of the embryo, rather than a specific cellular function revealed by GO analysis. To make this point more clearly in the manuscript, we have now added an explanatory sentence into the results "Grouping developmentally-expressed genes into sets based on their temporal expression profiles reduces complexity of analysis and can help distinguish signal from noise".

We agree that some readers will also wish to know about functional categories, and in response to this suggestion we have now added GO term enrichment into the manuscript. The full GO term enrichment analysis is reported in the revised Electronic Supplementary Information 1D and 1E; these data are now referred to in Results page 11.

2. the biological significance of the cellular system used should be discussed. The results obtained in this context may not reflect a good picture of the regulatory networks, which CRX/ETCH box genes may control in vivo and "function" in this context has a very specific meaning that should be explicit.

This is a very helpful suggestion. In this and previous work, we have found that ectopic expression of homeobox genes in embryonic fibroblasts (or indeed in adult cells) can recapitulate transcriptomic responses that occur in normal early embryonic development. Other have found the same, for example in studies of Dux genes. But it is true that this is not proof of 'function', rather it leads to the basis from which we can propose probable function.

We have therefore made some changes to wording to be explicit about this: **Abstract:** We state more clearly that ectopic expression is in cultured cells, and we change a ‘conclusion’ about function to a statement that our data are ‘consistent with’ particular roles.

Introduction: In the final paragraph, we insert the word ‘possible’ before function, we change ‘functional similarities’ to ‘similarities in activity’, and we change ‘indicative of subfunctionalisation’ to ‘consistent with subfunctionalisation’. **Results:** We insert a sentence to explain the biological significance of the cellular system: “Although this experiment does not mimic closely the in vivo situation, previous studies have found that ectopic expression of related homeobox genes in cultured cells can drive biologically-relevant transcriptomic changes ^{5,9}” **Discussion:** We insert the words ‘our data argue that’ in front of the marsupial CRX pleiotropy inference instead of treating this as a definitive conclusion. We have also changed the word ‘conclude’ to ‘propose’ in our explanation of DDC.

3. along the same line, the authors discuss ETCH box genes functions during early development but in the absence of direct functional data, based for instance of loss-of-function experiments in the mouse, this discussion remains extremely speculative. What is the direct evidence that "eutherians use multiple CRX-derived genes in early development when metatherian development can occur with a single CRX orthologue" (line 336)?

The published evidence for embryonic function is much stronger than ‘speculation’, but it does not include loss-of-function data. The embryonic functions of these genes have been inferred, in many published studies, from (a) transcriptomic effects after ectopic expression, in cellular systems ranging from ES cells (e.g. ref. 10) through to adult fibroblasts (e.g. ref. 5), (b) gene expression patterns conserved through evolution (which is a natural experiment; ref 5, 9, 11), and (c) molecular evolutionary patterns indicative of natural selection (ref. 5 & Katayama Sci Rep 8:17421). We consider these, collectively, to constitute strong indication of function. To increase clarity in the manuscript, as requested, we have modified the clause mentioned to “eutherians deploy multiple *CRX* -derived genes in early development when metatherian development occurs with expression of a single *CRX* orthologue”. This wording stresses activity rather than function.

4. The multiple losses undergone by these genes in eutherians should also be discussed.

Thank you for noting this omission. We have now added mention of this issue. In the Introduction, we add “There is also variation between eutherian species as to which ETCHbox are retained or lost, and which have been duplicated further.” In the Discussion, we refer back to this with possible explanations “The variation in copy number between eutherian species might also relate to variation in placental structure, or it may be an evolutionary consequence of partial redundancy ⁹”.

5. A clearer distinction between hypotheses, their rationale, and demonstrated facts would be needed.

We hope that we have achieved this by the many changes made in response to point 2, above.

Minor point: there are a lot of typos throughout the main text and Supplementary legends.

Those found have been fixed.

Referee: 2

Comments to the Author(s)

“The other gene copies underwent extreme sequence change, and are barely recognisable as duplicates of CRX. These are the genes that inherited the embryonic functions from CRX in ancestral mammal.” Please comment more on this apparent paradox.

We agree it is an apparent paradox. Interestingly, *ETCHbox* gene sequences are highly variable (fast-evolving) between species, as well as distinct from CRX, yet ectopic expression experiments suggest comparable functions between some of the genes and some species. We do not have a full explanation for this, but it may indicate that some activities are effected through partner proteins binding to small motifs, or it may indicate that it was the ancestral eye functions (not the embryo functions) that were more constraining of sequence evolution. This is speculation, and resolution of this paradox is a topic for future research. We have added a sentence into the discussion pointing out the interesting issue.

Metatherian mammals invented viviparity and one would think that whatever gene regulatory networks made such an evolutionary and revolutionary invention possible are likely to be conserved. For a whole other set of genes (albeit derived by duplication) to take over THE SAME invention requires some more persuasion than just “inherited.” As is, your description sounds as if viviparity were invented twice, once by metatherians and later by eutherians.

Apologies for not being clear enough. We do not suggest that viviparity evolved twice. It evolved once, but it has been modified in eutherians through evolution of a more invasive placenta and the evolution of embryo compaction and an inner cell mass (see Discussion). We do not suggest that a new set of genes took over functions; daughter genes are expected to have the same expression and function, hence we used the word ‘inherited’. To be clearer, we have now changed this word to ‘retained’ and explained more clearly how this relates to gene duplication: “We propose that metatherian *CRX*, and by extrapolation *CRX* of the common therian ancestor, functioned in the early embryo and the eye; following tandem gene duplication in eutherians, eye-associated functions were retained by one gene and embryonic functions were retained by a larger array of duplicates following DDC.”